# The Rare Marine Bioactive Compounds in Neurological Disorders and Diseases: Is the Blood-Brain Barrier an Obstacle or a Target?

**DOI:** 10.3390/md21070406

**Published:** 2023-07-18

**Authors:** Xiaozhen Diao, Hui Han, Bailin Li, Zhen Guo, Jun Fu, Wenhui Wu

**Affiliations:** 1Department of Marine Bio-Pharmacology, College of Food Science and Technology, Shanghai Ocean University, Shanghai 201306, China; xzdiao@shou.edu.cn (X.D.); m220300899@st.shou.edu.cn (H.H.); blli@shou.edu.cn (B.L.); 2Innovation Center, Shanghai BociMed Pharmaceutical Co., Ltd., Shanghai 201203, China; zhen.guo@bocimed.com (Z.G.); jun.fu@bocimed.com (J.F.)

**Keywords:** blood-brain barrier, marine bioactive compounds, tight junctions, transporters, aquaporin-4

## Abstract

The blood-brain barrier (BBB) is a dynamic barrier separating neurocytes and brain tissues from blood that is extremely sealed and strictly regulated by transporters such as aquaporin-4 (AQP-4), glucose transporter (GLUT), and specialized tight junctional complexes (TJCs) including tight junctions (TJs), adherens junctions (AJs), and Zonulae occludens (ZOs). With specifically selective transcellular and paracellular permeability, the BBB maintains a homeostatic microenvironment to protect the central nervous system (CNS). In recent years, increasing attention has been paied to the importance of BBB disruption and dysfunction in the pathology of neurological disorders and diseases, such as Alzheimer’s diseases (AD), Parkinson diseases (PD), stroke and cerebral edema. However, the further research on how the integral structure and function of BBB are altered under the physiological or pathological conditions is still needed. Focusing on the ultrastructural features of the BBB and combining the latest research on associated proteins and transporters, physiological regulation and pathological change of the BBB were elucidated. By summarizing the protective effects of known bioactive compounds derived from marine life on the BBB, this review aims to highlight the BBB as a key to the treatment of several major neurological diseases instead of a normally described obstacle to drug absorption and transport. Overall, the BBB’s morphological characteristics and physiological function and their regulation provide the theoretical basis for the study on the BBB and inspire the diagnosis of and therapy for neurological diseases.

## 1. Introduction

So far, the BBB is well-known as a firmly guarded “gate” of the cerebral microenvironment against not only pathogens but also drugs or bioactive molecules, in which case great efforts have been devoted to drug delivery strategies to improve their permeability through the BBB [1]. However, its role as a therapeutic target of neurological disorders and diseases came into our horizon with the further investigation into structural or functional proteins of the BBB, especially the discovery of AQP-4 as a membrane “water channel” by Peter Agre et al., rewarded in 2003 with the Nobel Prize. Evidence of the correlation between the BBB disruption or dysfunction and CNS diseases also supports the perspective [2,3,4]. Although it seems to be a chance for us to seek new therapeutic strategies and develop novel drugs or bioactive compounds, limited achievements have been found until now. Moreover, due to their high safety and bioavailability compared to chemical drugs, natural bioactive compounds are among the most sought-after spots in the treatment of CNS diseases. Therefore, inspired by previous research on the structural features and functional characteristics of the BBB [5], compounds targeting the BBB with promising neuroprotective effects, especially those derived from marine life, were discussed with the underlying mechanisms based on the physiological regulation and pathological change of the BBB in this review.

## 2. Ultrastructural Features of the BBB

The BBB is composed of three spatial membranes spatially from the capillary to the parenchyma, which include: (a) the endothelial membrane constructed by the cytoplasmic membranes and tight junctional complexes (TJCs) between brain microvascular endothelial cells (BMECs), (b) the basement membrane formed by a network of extracellular matrix proteins secreted by BMECs, pericytes, and glial endfeet, and (c) the glia limitans formed by the endfeet of astrocytes (ACs) surrounding the capillary [6] (Shown in Figure 1). These membranes are correlated by the coordinated communication networks so-called neurovascular units (NVUs), which are composed of endothelial cells and other CNS cell types (such as pericytes, astrocytes, microglia, and leukocytes) to form barrier characteristics and function together [7].

### 2.1. BMECs and BBB Tight Junctional Complexes

The junctional structures between BMECs form a physical barrier to provide highly selective permeability for the ions and molecules to maintain the homeostasis of the neural microenvironment and resist invasion and attack by endogenous or exogenous pathogens, which are known as tight junctional complexes (TJCs). TJCs fill up the intercellular interval between adjacent cells by TJs, AJs, and ZOs to construct the adjustable channels which only allow water or ions to pass through [6]. TJs consist of over forty kinds of proteins, mainly including claudins (CLDNs), occludin (OCLN), and junctional adhesion molecules (JAMs) [4]. So far, there are twenty known isoforms of CLDNs (CLDN-1~20) but only four of them (CLDN-1, 3, 5, 12) have been identified in the BBB [8], among which CLDN-3 and CLDN-5 own the highest expressions [9,10,11] and play vital roles in maintaining the TJs integrity [8]. Each CLDN monomer comprises four transmembrane helical bundles and forms two extracellular loops by folding, which are the ECL1 (determines structural compactness and selective permeability) and ECL2 (involved in intermolecular interactions) domains (Shown in Figure 2) [12]. The expressions of OCLN and JAMs (only JAM-1 and 3 are expressed in the BBB) also influence the structure and function of the TJs [13]. AJs beneath TJs construct an adhesion belt across the intercellular crevice by vascular endothelial cadherin (VE-cadherin), connecting to actin cytoskeleton via catenins (α, β and γ) to form the E-cad/cat complex which provides cell-cell adhesion and structural support for TJs [14]. Although there has been no evidence for its determination of paracellular permeability so far, AJs’ structural destruction contributes to the disruption of TJs [15]. ZOs, mainly including ZO-1/2/3, link the TJs’ proteins to intracellular actin and cytoskeleton by cingulin, which play an important role to further inhibit the penetration of polar solutes from the plasma into the cerebral extracellular fluid via intercellular diffusion [16].

### 2.2. Astrocytes and Aquaporin-4

The surrounding astrocytes extend their neurites to form perivascular end-feet that attach to the surface of capillaries [18]. Astrocytes are highly polarized cells which express various transport proteins such as aquaporin-4 (AQP-4), glucose transported type 1(GLUT1), big current potassium (BK) channels [19,20]. Therefore, astrocytes not only maintain BBB function but also regulate cerebral blood flow to support neuronal metabolism in bidirectional neurovascular coupling, which is another essential participant in the regulation of the microenvironment in the central nervous circulatory system (CNS), along with the BBB [21]. Furthermore, astrocytes participate in the clearance of the wastes produced from brain [22]. AQP-4 which is highly concentrated on the cell surfaces of the astrocytic endfeet play a role in the maintenance of cerebral water homeostasis and neural conduction, and it is implicated in the development and resolution of edema in the pathophysiology of stroke [23]. Each monomer of AQP-4 existing as tetramers with four independent pores is composed of two similar halves integrating in opposite orientations by their internal pseudo two-fold symmetry [24]. (Shown in Figure 3).

### 2.3. Basement Membrane

The basement membrane, including the endothelial basement membrane and parenchymal basement membrane, is composed of laminins, collagen IV, nidogens and heparan sulfate proteoglycans [25,26]. The main functions of the basement membrane include: (1) supporting and facilitating the cell-cell interactions in the BBB, (2) impacting on BBB function by anchoring cellular components to the barrier via the interactions with integrin and dystroglycan receptors, and (3) participating in CNS immune privilege [27]. However, such an important barrier is hard to be studied apart from the entire BBB due to the failure of the development past embryonic stage in knockout models and the difficulty in rebuilding the membrane in vitro without the non-negligible interaction with NYU cells [28].

### 2.4. Pericytes

Pericytes are motile cells embedded in the basement membrane, which are physically attached to BMECs [29]. Pericytes function in keeping BBB integrity, especially as shown in neurological diseases, which indicates its physiological importance [30]. Its newly found function in giving the direct contractile force on BMECs to actively modulate the microvascular tone remain argued [31], which should be further testified by using the improved imaging methods with more specific markers and higher resolution to erase the ambiguous distinctions between pericytes and smooth muscle cells [27].

Except for the cells described above, microglial and immune cells also influence the BBB by secreting inflammatory mediators and matrix metalloproteinases (MMPs), both of which induce BBB disruption or dysfunction [32,33].

## 3. Regulation of the BBB

The BBB is a dynamic system regulated by a series of cellular interactions rather than a simple physical wall, whose physiological properties can be influenced by the factors including changing the neural microenvironment, aging, high-fat diet, CNS inflammation and diseases [34]. The regulation of the BBB under physiological and pathological conditions mainly lies in the following aspects: (1) cell-cell interaction within NVU cells (such as pericytes, astrocytes, microglia, and leukocytes), (2) paracellular transport by TJCs proteins (such as CLDNs, ZOs, OCLN, etc.), and (3) transcellular transport by BBB transporter and receptor proteins (such as AQP-4, GLUT-1, and Mfsd2a). (Shown in Figure 4).

### 3.1. Surrounding Neural Microenvironment

Transplanted avascular brain tissue that induces BBB-like properties of gut endothelium in chick embryos was used to prove the fact that the signals from the neural microenvironment induce BBB formation in the development, but the dorsal mesoderm transplanted into the embryonic brain failed to help the vessels develop barrier properties, which emphasizes the necessity of brain BMECs in BBB formation [35]. BBB regulation is a complex progress involving all kinds of NVU cells, none of which can be isolated or underestimated.

Supported by their initial postnatally appearance [36], astrocytes are preferably to be considered to play a role in BBB maintenance rather than its formation, whose secreted factors or proteins such as Sonic Hedgehog [37], angiotensin [38,39], and phospholipid transporter molecule apolipoprotein E (APOE) [40] mediate the regulatory processes of BBB tightness. 

During early angiogenesis and BBB formation, pericyte recruitment promoted by platelet-derived growth factor (PDGFβ) is a critical step in the vessels’ development [41], which is followed by further communication between pericytes and cerebral BMECs by transforming growth factor β (TGFβ) and its receptor (TGFRβ) [42]. Unlike astrocytes, pericytes not only participate in barrier differentiation in the development but also influence BBB integrity via the effect on TJs expression in adulthood, according to recent research [43]. This study also found the inducement of the localization onto the cerebral vessel wall and the polarization of astrocyte endfeet by pericytes, which implied close correlation among NVU cells.

Other cells in the neural microenvironment such as microglia, and leukocytes also have been proven to take part in BBB regulation. For instance, microglia cells take different roles according to their active states, including the inducement of BBB damage by releasing proinflammatory cytokines (ILK-1β, TNF-α) via a classic activated M1 pathway, and the support on tissue repair by releasing chemokines, vascular endothelial growth factor (VEGF) and activating the neurotrophic pathway via the alternative M2 pathway [44,45]; Leukocytes (T cell, neutrophils) increase BBB permeability by releasing cytokines and ROS mediators [46].

### 3.2. Paracellular Permeability Regulated by TJCs

The fundamental function of the BBB is shown by its specifically selective and extremely strictly regulated paracellular permeability whose change largely reflects BBB structural integrity and function. Paracellular passages of small molecules and nonpolar substance from blood to brain are mediated by specialized TJs and other TJCs proteins, whose structures and expressions are essential to BBB regulation.

As mentioned above, TJs are composed of several kinds of transmembrane proteins that support and interact with each other. Although there are still some blank space and obscure parts about how specific protein acts for TJs construction and functioning, enough evidence has been given to prove their importance for TJs integrity and BBB formation. Among all TJs proteins, CLDNs (CLDN-1 and 5) claim to be the main functional and integral proteins of TJs and key components of BBB regulation [17]. OCLN is another regulatory protein, which continuously distributes and highly enriches in the brain more than other nonneural tissues [47]. Although OCLN-deficient mice developed normal TJs [48], OCLN absence induced the mislocalization of tricellulin proteins from tricellular TJs to bicellular TJs [49], which provide potential correlation between OCLN and BBB function. This is supported by recent research on Lsr’s (a tricellulin protein) role in BBB formation [50]. JAMs function in cell-cell adhesion to strengthen the connection between adjacent BMECs, but their further functions in barrier function are still unrevealed. As cytoplasmic accessory proteins, ZOs family proteins contain various domains, which provide the binding sites for TJs and AJs proteins. As reported, the PDZ1 domain of ZO-1~3 bind with CLDNs at their COOH-terminal [16], and the COOH-terminal of ZO-1 and -2 are bond to the primary cytoskeleton proteins [51]. Also, it has been recently reported that the GUK domain of ZO-1interact with OCLN [52], and JAMs bind directly to ZO-1 [53]. In the case of BMECs in the BBB, mainly ZO-1 and ZO-2 provide support for TJs structure and are important to BBB regulation [8]. There are many interactions between AJs and TJs and ZOs during the assembly of junctional structures. Except for its function in structural binding, VE-cadherin was reported to increase CLDN-5 expression and localization in the BBB [54]. Furthermore, the junction-associated signalling system Wnt-β-catenin pathway based on β-catenin contributes to both TJs formation during the embryonic stage and barrier maintenance in the maturation stage of the BBB by promoting BBB gene expression [55].

BBB paracellular permeability can be influenced by not only pathological conditions but also physiological inputs including hormones, cytokines, Na^+^-coupled solute transport, myosin activity, and cell signaling pathways [56]. So far, many factors including MMPs-dependent degradation, phosphorylation, ubiquitination and cytokines (TNF-α, IL-1 β, IFN-γ, HGF) have been proven to regulate OCLN in the BBB. MMP-2 and -9 mediate OCLN degradation by their activation and are mediated by different pathways under pathological conditions, supported by evidence in vitro and in vivo [57,58]. OCLN contains multiple phosphorylation sites, in which case, OCLN phosphorylation shows different influences on paracellular permeability depending on its phosphorylation type and the mediating signaling pathway [59]. In eukaryotic cells, ubiquitin leads OCLN to the proteasome degradation to disrupt the BBB [60], which may also induce CLDNs and ZOs endocytosis [61]. There are two main aspects from which CLDNs expressions are regulated. One is from the differentiation-specific gene expression aspect, in which CLDNs’ expressions can be affected on the transcription or translation level. For example, the snail family transcriptional repressor (Snail) triggering the epithelial-to-mesenchymal transition (EMT) decreases CLDN-3, -4, and -7 expressions [62] and CLDN-1 expression [63] at transcription and translation levels separately. Other transcription factors such as GATA-4 were reported to show the regulation on specific CLDNs [64]. Another is the acute change caused by hormone, cytokine or other factors according to extensive evidence [65,66]. Otherwise, there have been some new ways of acute regulation of CLDNs attracting great interest recently including posttranslational modifications in phosphorylation and the endocytosis of CLDNs [67].

### 3.3. Water Transport Mediated by AQP-4

As the main water channel in CNS, AQP-4 mediate water transport in bidirectionally through cells to maintain brain osmolarity and regulate excitability [68]. Although there has not been much research on the regulation of AQP-4 directly related to BBB physiology so far, its role in the pathogenesis of cerebral edema along with BBB dysfunction and disruption has been examined by multiple mouse models. Due to its characteristic of bidirectional water transport, AQP-4 has the opposite function in cerebral edema via different pathomechanisms [23]. In a cytotoxic brain edema induced by cerebral ischemia and traumatic brain injury, AQP-4 facilitates water-entering into astrocytes to form edema. Compression of the capillary by swollen astrocyte end-feet cause BBB disruption, which eventually leads to the leakage of RBCs. In contrast, in a vasogenic edema induced by brain tumor and persistent ischemia, AQP-4 helps with the clearance of vasogenic cerebral edema fluid due to BBB leakage. However, the process in cycle is much more complicated to be clarified due to the shifting from cytotoxic to vasogenic edema and hemorrhagic conversion during the pathomechanism of cerebral edema.

### 3.4. Transporters in BBB

Influx and efflux transporters are expressed on the surface of cerebral BMECs in BBB, some of which have double functions in both mediating the transport of nutrients, and regulating BBB integrity [69]. The glucose transporter GLUT-1, a solute carrier transporters (SLCs), encoded by Slc2A1 gene, shows the potential impacts on BBB integrity and its physiology, which was indicated by the BBB disruption prior to AD pathology in GLUT-1 deficient AD mice [70]. Mfsd2a, the receptor for ω-3 has been recently proven to play a critical role in BBB formation and integrity, which was supported by the BBB leakage in both the embryonic stage and adulthood of Mfsd2a-knockout mice [71]. The decreasing Mfsd2a level was also found in pericyte-deficient mice, which indicates that pericytes support BBB integrity via mediating Mfsd2a expression.

## 4. Marine Bioactive Compounds and BBB-Related Neurological Diseases

BBB is an extremely sealed and strictly regulated dynamic barrier between blood and the neurocytes or brain tissues with selective permeability [6]. The disruption or dysfunction of the BBB have been proven to be correlated with neurological disorders and diseases, such as Alzheimer’s diseases (AD), Parkinson diseases (PD), and stroke [72]. So far, many challenges remain to be overcome regarding the therapy strategies of these diseases, for instance, symptomatic effects rather than ideal cures, limited approval drugs with unavoidable side effects and degrading improvement effects, which require us to search for more natural bioactive compounds due to their high safety and bioavailability. Compounds derived from marine life have been under discussions here for their huge potential to not only provide positive effects on these diseases but also for possessing other bioactivities. 

### 4.1. AD 

The pathological study and the seeking of the treatment of neurodegenerative diseases induced by aging and lifestyle, especially AD is in the urge demand due to its increasing worldwide influence [73]. The amyloid plaques formed by aggregated beta-amyloid peptides (Aβ) outside and the neurofibrillary tangles (NFT) caused by tau protein phosphorylation inside the neurons are the main pathological phenomenon which gradually contribute to AD [74]. BBB disruption was indicated as the biomarker of cognitive impairment in the early stage of AD by an increased concentration of soluble plate-derived growth factor receptor-β (s PDGFR-β), which is the potential cerebral spinal fluid marker of BBB dysfunction [75]. Individuals at the genetic risk of AD carrying E4 variant of apolipoprotein E (APOE4) showed increased activity of the matrix metalloproteinase (MMP)-9 which induces BBB disruption [76]. Furthermore, BBB dysfunction as one of the pathophysiological changes caused by Aβ deposition through factors including MMPs, ROS, NF-KB, and Ca^2+^-CaN aggravates the pathological progress in AD mainly by triggering cognitive decline [77].

A limited amount of drugs has been approved by FDA currently, which mainly target the neurotransmitter, and these all have serious side effects with degrading improvements on the symptoms until 12 months after administration [78]. New drugs or supplements on the sustained improvement of the cognitive decline are in urgent demand. So far, multiple marine bioactive compounds have been found to be effective for the treatment of AD by reducing Aβ production and aggregation as well as by inhibiting AchE activity and inflammatory responses. Except for docosahexaenoic acid-containing phospholipids (DHA-PLs, Shown in Figure 5A) extracted from fish or krill oil and algae [79], fascaplysin (Shown in Figure 5B) as an alkaloid extracted from sponge, induces the activity of the extracerebral transporter protein (P-pg) and inhibits Aβ production and aggregation to alleviate AD symptoms [80]. AchE is a potential therapeutic target for the treatment of AD since Ach deficiency leads to cognitive decline in patients. Circumdantin D (Shown in Figure 5C), one of the coral extracts, shows a high inhibitory effect on AchE and can also increase cell viability, reduce apoptosis and interfere with pro-inflammatory response [81]. Other bioactive compounds including irene-carbolines A (or B) extracted from sea squirt and petrosamine extracted from sponge (Shown in Figure 5D–F), have the same inhibitory effects with similar underlying mechanisms [82].

### 4.2. PD

PD is the most common neurodegenerative diseases second to AD, whose neuropathological features are the loss of nigral dopaminergic neurons and the accumulation of Lexy bodies caused by α-synuclein (α-syn) protein aggregation. The BBB dysfunction allows environmentally-derived toxins to enter the brain when exposed to the exogenous risk of PD, which indirectly leads to its involvement in PD pathogenesis [82]. The potential correlation between PD and BBB disruption was indicated by oxidative stress [83], neuroinflammation [84], and angiogenesis [85]. Multiple marine bioactive compounds have been proven to prevent or treat PD by reducing the abnormal aggregation of α-syn, alleviating mitochondrial dysfunction, and inhibiting neuroinflammation or oxidative stress (OS). Along with its anti-inflammatory effects, the soft coral extract 11-Dehydrosinulariolide (Shown in Figure 6A) can also protect neurons by reducing 6-hydroxydopamine-mediating cytotoxicity and apoptosis mainly through the up-regulation of PI3-K/Akt signaling [86]. Ethyl acetate derivatives (Shown in Figure 6B) derived from sea cucumber share the same mechanism with 11-Dehydrosinulariolide to alleviate PD, which also involve its inhibiting α-synuclein aggregation [87]. Being a precursor of uric acid (UA, an antioxidant), inosine (Shown in Figure 6C) isolated from sponges could alleviate PD symptoms through its long-term use via increasing UA concentrations in serum and cerebrospinal fluid to inhibit OS [87]. Likewise, astaxanthin (ATX) (Shown in Figure 6D), an antioxidant extracted from algae, was proven to inhibit the activation of microglia and the release of pro-inflammatory cytokines in the brain [88]. Furthermore, both ω-3 fatty acids from fish or algae and Pramipexole (Shown in Figure 6E,F) from marine yeast were proven to improve depressive symptoms caused by dopamine dysfunction in PD patients [87,89]. 

Autophagy as a conserved process in the evolution of eukaryotic cells takes vital responsibility in cell proliferation, death, and its defense against infections, aging, or various diseases, which is mainly achieved by (1) the degradation of aged or defective proteins, macromolecular complexes, and organelles, (2) the clearance of invaded microbiota or aggregates of toxin proteins [90]. The neurons’ death and the failure in guarantying BBB integrity and maintaining microenvironment homeostasis, which are caused by impaired autophagy in PD pathogenesis which, in turn, aggravate symptoms and accelerate progress. So far, various regulators involved in autolysosome (ALP) pathways to promote α-syn degradation by activating autophagy, such as 5′AMP-activated protein kinase (AMP), mammalian target of rapamycin 1 (mTORC1), UNC-51-like kinase 1 (ULK1), IMPase, LRRK2, beclin-1, transcription factor EB (TFEb), GCase, estrogen-related receptor (ERRα), and c-Abelson (c-ABL) have been proven to be targets and their activators or inhibitors have provided promising use in PD treatment [91]. Although lack of organ specificity and substrate selectivity of these autophagy-associated therapy strategies could be respectively overcome by the development of LRRK2 and the targeting of chaperone-mediated autophagy (CMA) [92,93], there are remaining problems to be solved. What would happen to the complete BBB integrity and barrier function with a change in the microenvironment induced by the degradation of α-syn and other proteins or organelles? What if neurons suffered the environmental stress from the extra degradation products by excessive autophagy? How dose autophagy interact with other intracellular activities in such a complicated system as the BBB and what is the consequent change in the microenvironment? Further investigation on autophagy, along with the BBB in PD pathology, is demanded.

### 4.3. Stroke 

Stroke is the leading cause of death worldwide, second only to ischemic heart disease, which is categorized as hemorrhagic or ischemic according to their pathomechanism [94]. Although hemorrhagic and ischemic strokes cause cerebral edema (defined as vasogenic and cytotoxic edemas respectively) formation through different pathways, they undergo an advancing and continuously reciprocal conversion with sharing the common facilitating proteins, such as tight junctions (TJs), aquaporin-4, and other factors [23]. Once the perifocal edema occurs following initial cerebral trauma in a hemorrhagic stroke, clot-derived proteins or vasoactive substances contribute to BBB impairment leading to the formation of membrane attack complexes and the destruction of the CNS and RECs [95]. The lysis of RECs contains hemoglobin that causes so-called ‘hemoglobin toxicity’, which induces cellular damage [96], after which the secondary damage conjointlywith the cytotoxic edema are facilitated by inflammatory mediators, interleukins, and MMPs [97]. In an ischemic stroke, thrombus formation occludes the cerebral blood flow leading to the functional insufficiency of Na^+^/K^+^ ATPnase caused by impairing the ATP synthesis, which consequently contribute to the intracellular Na^+^ accumulation. Together with the lactate produced by anaerobic glycolysis, excessive intracellular Na^+^ impels water into the cells to form cytotoxic edema [98]. BBB disruption is caused by further cellular damage through several mechanisms and factors including reverse pinocytosis [99], VEGF [100], and MMPs [101], during which the ischemic nidus undergoes hemorrhagic conversion. As concluded, BBB dysfunction follows a biphasic time course during a stoke.

In all, not only can BBB disruption or dysfunction lead to stroke pathology, but also inflammation and mitochondrial disorders. Due to the complicated pathological progress of a stroke, the most known bioactive compounds show comprehensive effects involving multiply factors mentioned above to influence the symptoms and development of a stroke. Xyloketal B (Shown in Figure 7A), isolated from *Xylaria* sp., was suggested to be applied in the therapy of an ischemic stroke by inhibiting BBB disruption, and alleviating mitochondrial disorders caused by excessive ROS, along with its anti-inflammation or anti-apoptosis effects [102]. As an isoindolone alkaloid from marine fungi *Starchbotrys longispora* FG216 and *Stachybotrys microspora* IFO 30018, and, apart from its thrombolysis activity, SMTP-7 (also FGFC1) (Shown in Figure 7B) was proven to alleviate cerebral injury by inhibiting OS and the secretion of proinflammatory or inflammatory cytokines [103]. Tramiprosate (Shown in Figure 7C) extracted from red algae produced a neural protective effect by disrupting the interaction between postsynaptic protein-95 (PSD-95) and nitric oxidase (nNOS) and by inhibiting the translocation of nNOS to the plasmic membrane [104]. NP04634 (Shown in Figure 7D), a marine-origin novel compound with cytoprotective properties was indicated to have potential therapeutic implications in a stroke involving mitochondrial disruption [105].

As the largest ecosystem in the Earth biosphere breeding numerous bio-resources, marine should have provided more bioactive compounds than terrene, which is the opposite to the current situation. Meanwhile, although it has been strongly suggested by previous studies that compromised BBB are associated with various neurodegenerative disorders and cerebrovascular disturbances, their exact roles in the pathological progress remain ambiguous. The implication of BBB dysfunction in the pathogenesis, etiology, and treatment have been also underestimated. Therefore, there is still much to be investigated on the integral structure and function of the BBB under the physiological or pathological conditions through its key proteins or factors, along with the continuous exploration on potential marine bioactive compounds.

## 5. Conclusions

The morphological characteristics of the BBB are determined by its ultrastructure which is mainly constructed by TJCs between BMECs and supported by pericytes and astrocytes. Unlike the other barriers in human body, the BBB’s physiological function is also achieved by a series of complex cell-cell interactions involved by BMECs, astrocytes, pericytes and neurons, except for the paracellular passage regulated by TJs and transcellular transportation facilitated by various transporters. The integrity and function of the BBB are guaranteed by these junctional proteins and transporters along with the maintenance of the surrounding microenvironment, which could be regulated under the physiological and pathological conditions. Recently, the BBB as a therapeutic target rather than a barrier to drugs or bioactive compounds has attracted increasing attention with continuously new findings on its key mediating factors of structure and function and the underlying mechanisms. 

## 6. Future Perspectives

Although possessing less side-effects than CNS chemical drugs, natural compounds commonly with a large size and polar nature are difficult to cross the BBB, for which huge efforts with numerous resources and countless funds have been made to improve their passage through the barrier in decades but with relatively slow progress. Except for the lack of BBB permeability and limited bioavailability of most bioactive molecules, nonspecific targeting also limits their application in therapeutic strategies, which could be remedied by their influences on specific proteins or transporters of the BBB itself. Furthermore, differences in neuroprotective effects between preclinical and clinical trials of many promising compounds in preventing and treating degenerative diseases or disorders may indicate their complicated interaction with the proteins or transporters while they are transported through the BBB. Therefore, instead of commonly being an intractable obstacle that drugs or bioactive compounds must overcome during their absorption and metabolism in the brain, a perspective that the BBB itself could be a therapeutic target of these drugs or compounds, which deserves more attention on its ultrastructure and function is proposed here. 

All the research findings and latest discoveries on the maintenance and regulation of BBB ultrastructure and normal function inspire the diagnosis and therapy in neurological diseases.

## Figures and Tables

**Figure 1 marinedrugs-21-00406-f001:**
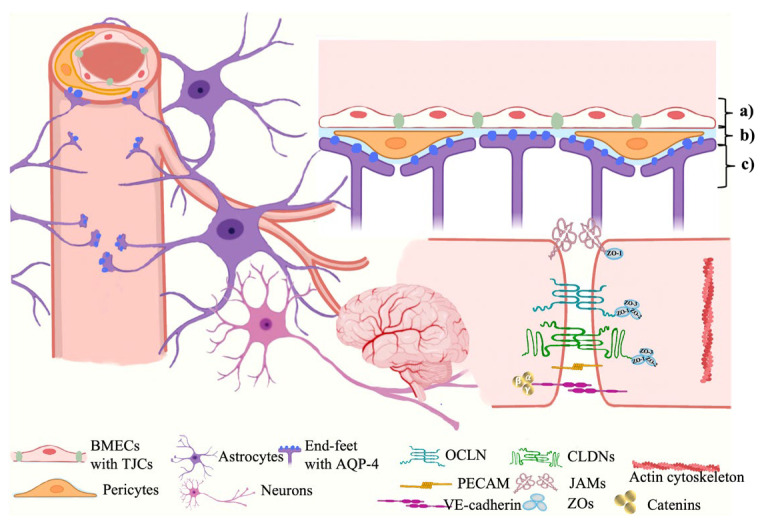
The BBB ultrastructure. The BBB is mainly constructed by three different membranes including (a) the endothelial membrane, (b) the basement membrane, and (c) the glia limitans, which involves various neurovascular unit (NVU) cells, structural and functional proteins of TJCs, and transporters, etc. BMECs, brain microvascular endothelial cells; TJCs, tight junctional complexes; AQP-4, aquaporin-4; OCLN, occludin; CLDNs, claudins; PECAM, platelet endothelial cell adhesion molecule; JAMs, junctional adhesion molecules; ZOs, zonulae occludens.

**Figure 2 marinedrugs-21-00406-f002:**
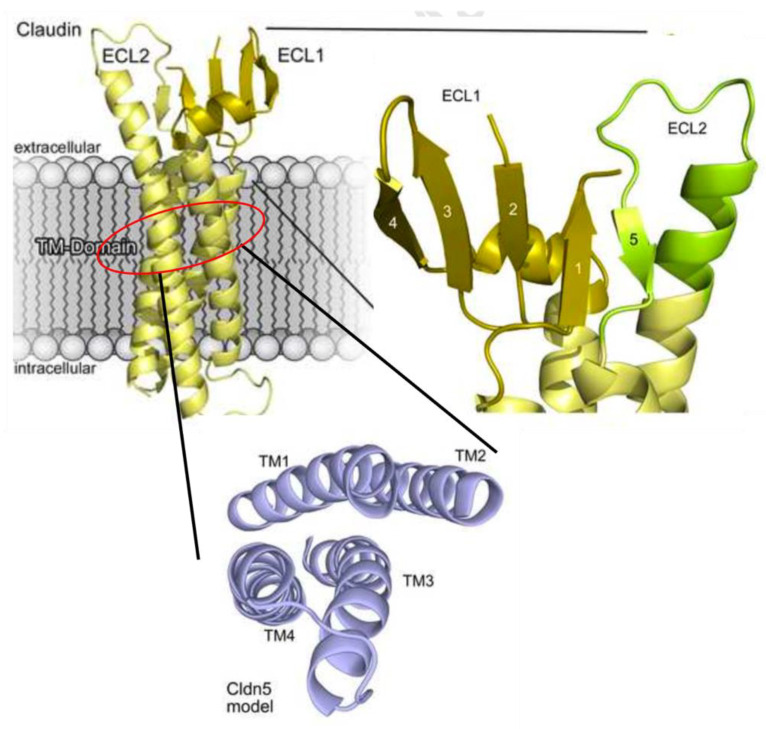
Protein model of CLDN5. The tertiary structure of CLDN5 monomer mainly consist of the extracellular loop 1 and 2 (ECL1 and ECL2) domains, which are formed by the folding of four transmembrane helical bundles. (Quoted from G. Krause, et al. [17]).

**Figure 3 marinedrugs-21-00406-f003:**
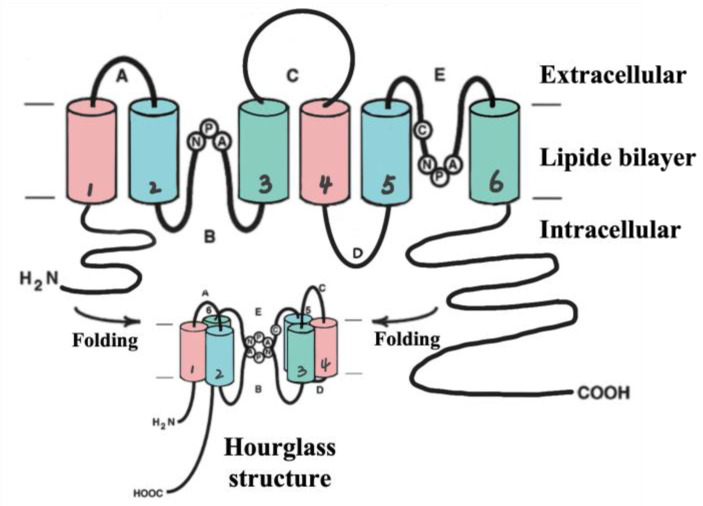
AQP-4 structure (hourglass structure). AQP-4 is comprised of four subunits (~28 kDa), each of which contains six transmembrane domains including helix 1~6 (H1~6). Loop B (between H2 and H3) and loop E (between H2 and H3) contribute to the hourglass structure after folding to provide the water channel.

**Figure 4 marinedrugs-21-00406-f004:**
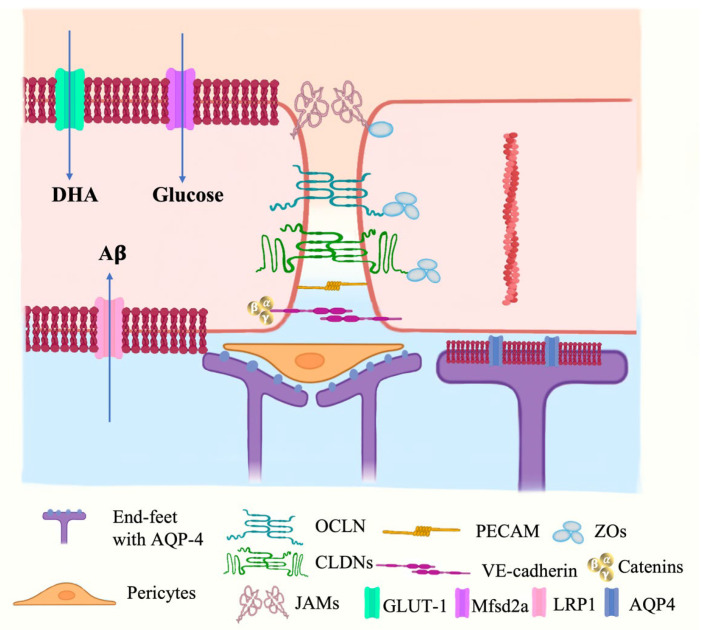
Junctional proteins and transports in the BBB microenvironment. Neural microenvironment cooperating with the interactions among NVU cells and transportation mediated by TJCs proteins or transporters impact on BBB regulation together. DHA, docosahexaenoic acid; Aβ, beta-amyloid peptides; GLUT-1, glucose transporter-1; Mfsd2a, major facilitator superfamily domain containing 2a; LRP1, lipoprotein receptor-related protein 1.

**Figure 5 marinedrugs-21-00406-f005:**
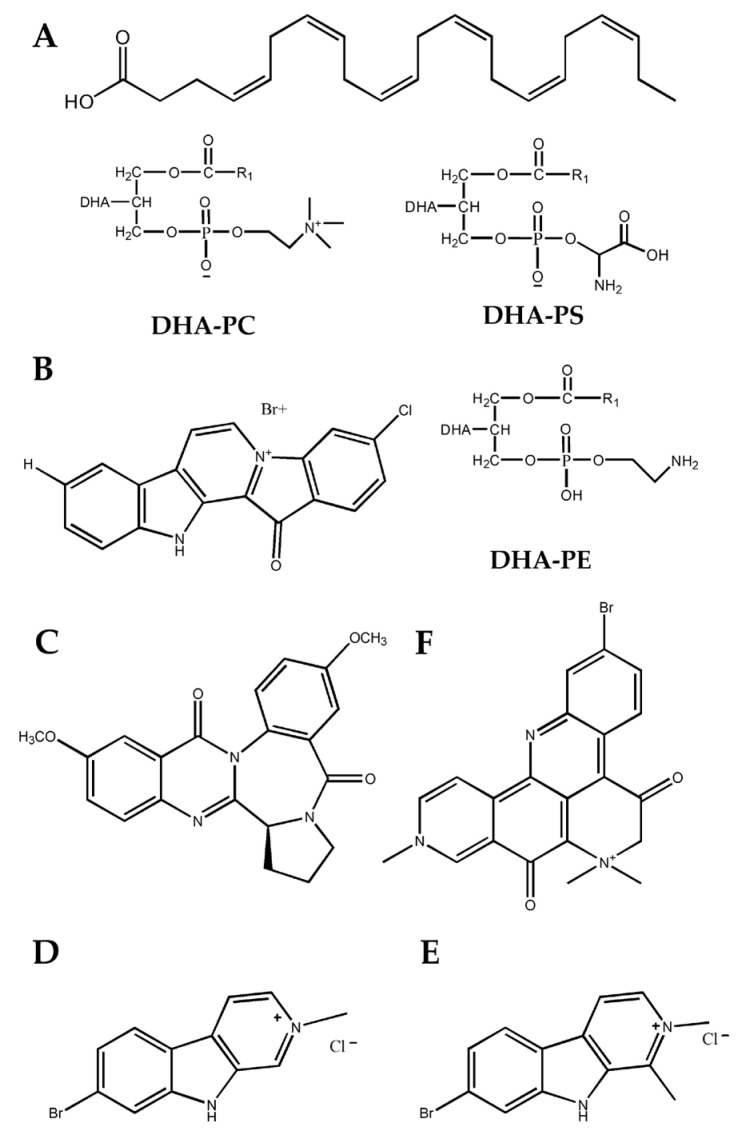
Marine bioactive compounds effective for treating AD. Bioactive compounds derived from different marine resources including docosahexaenoic acid-containing phospholipids (DHA-PLs) (**A**), fascaplysin (**B**), circumdantin D (**C**), irene-carbolines A or B (**D**,**E**), and petrosamine (**F**) have been proven to reduce Aβ production and aggregation, as well as inhibit AchE activity or inflammatory responses. DHA-PC, DHA-phosphatidylcholine; PS, DHA-phosphatidylserine; PE, DHA-phosphatidylethanolamine.

**Figure 6 marinedrugs-21-00406-f006:**
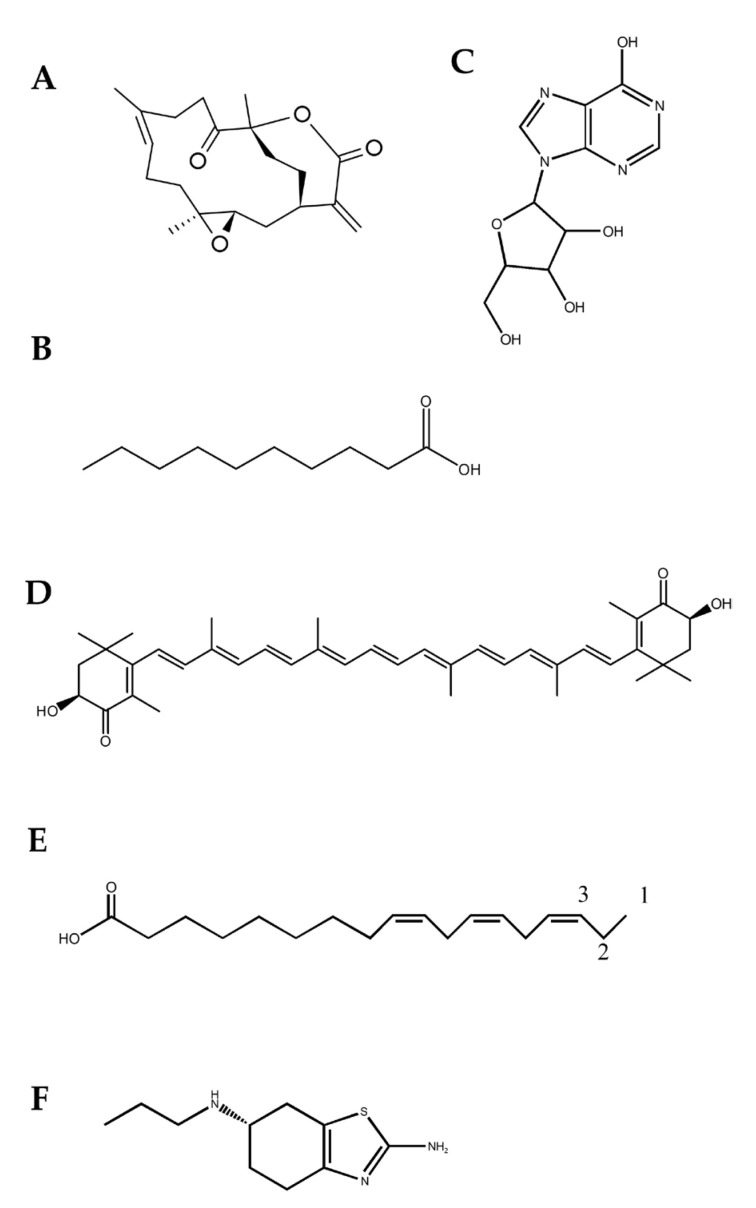
Marine bioactive compounds effective for treating PD. Bioactive compounds derived from different marine resources including 11-Dehydrosinulariolide (**A**), ethyl acetate fraction (HLEA) (**B**), inosine (**C**), astaxanthin (ATX) (**D**), ω-3 fatty acids (**E**), and pramipexole (**F**) have been proven to reduce abnormal aggregation of α-syn, alleviate mitochondrial dysfunction, and inhibit neuroinflammation or OS.

**Figure 7 marinedrugs-21-00406-f007:**
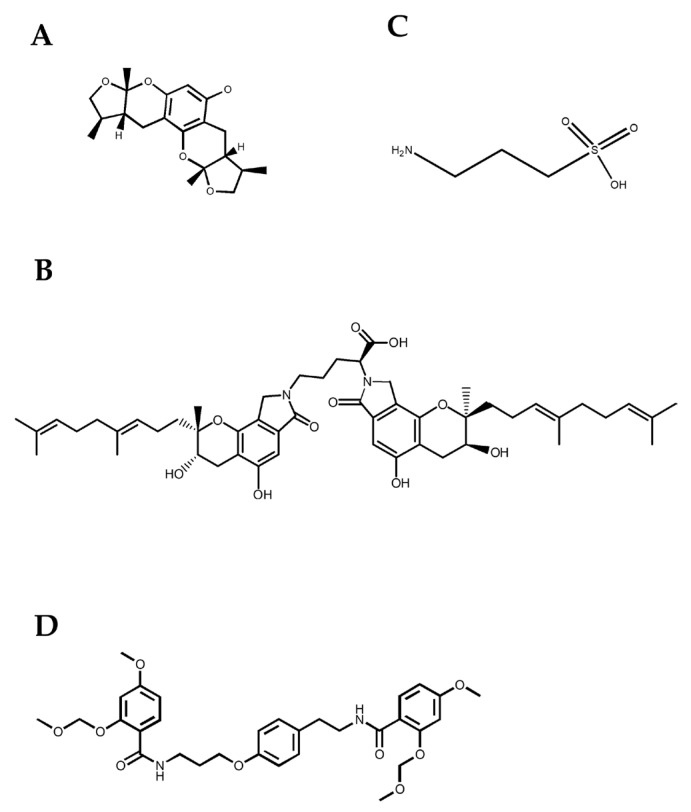
Marine bioactive compounds effective for treating stroke. Bioactive compounds derived from different marine resources including Xyloketal B (**A**), FGFC1 (**B**), Tramiprosate (**C**), and NP04634 (**D**) have been proven to inhibit BBB disruption and also to alleviate mitochondrial disorders, inflammation and apoptosis.

## Data Availability

No new data were created or analyzed in this study.

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
