# Peer review of "The Rare Marine Bioactive Compounds in Neurological Disorders and Diseases: Is the Blood-Brain Barrier an Obstacle or a Target?"

_marinedrugs, 2023, doi:10.3390/md21070406_

Round 1
Reviewer 1 Report
The manuscript is of interest, clearly presented and well written. The reviewer suggest to present a better coverage of the review with recent published papers and a better definition and additional references for the neurovascular unit (NVU).
Is figure 1 composed by the authors or resulted (adapted) from a previously paper? in this case is a reference available? Please also check all the following figures
Figure 5, detail the abbreviations in the legend to the figure
Reviewer 2 Report
This is a review of the blood-brain barrier (BBB) and its role in protecting the central nervous system, as well as the physiological regulation and pathological changes of BBB, and the protective effects of marine bioactive compounds. Overall, the review covered the topic in a reasonably comprehensive manner.
Given the question in the review title, the authors could have argued more explicitly why they consider the BBB to be a promising target rather than just an obstacle for marine compounds.
Gaps in knowledge should be highlighted more explicitly, particularly regarding the role of the BBB as an obstacle/target.
A separate "FUTURE PERSPECTIVES" section addressing the gaps mentioned in the review will make it easier for the reader to navigate and gain useful information from the review.
There is no INTRODUCTION section. Perhaps the authors could include an INTRODUCTION and provide some background information, e.g., the rationale for writing the review, how selected papers were chosen for discussion, etc. There are previous reviews (e.g., Archie, S.R.; Al Shoyaib, A.; Cucullo, L., Blood-brain barrier dysfunction in cns disorders and putative therapeutic targets: A review. Pharmaceutics 2021, 13, (11)). So, in the INTRODUCTION, the authors could highlight the novelty of their review in contrast to previously published reviews.
The ABSTRACT could be revised to include at least a brief statement addressing the question in the review title "Is the blood-brain barrier an obstacle or a target?”. As it is, the ABSTRACT seems somewhat disconnected from the review title.
There is only ONE reference published in 2023, which is too limited. Could the authors check this again?
In the ABSTRACT, the statement "Focusing on the ultrastructural features of the BBB... marine on BBB." is too long and confusing.
Figure 1 can be improved by giving the full abbreviation in the figure legend. To help readers understand the figure, please indicate in the figure the three different membranes (a-c) mentioned in the legend.
Figure 3 should be cited in the text before it is presented.
In line 118, shouldn’t “Fig. 2” be “Fig. 4” instead?
For Figures 5-7, the sizes of the structures/drawings do not seem to be standardized, e.g., some aromatic rings in the same figures are larger, some smaller. Some aromatic rings also seem to be distorted, e.g., in Figure 7a. Please check again.
The CONCLUSIONS look good and coherent. Here the authors have also addressed the question in the review title.
